# The Effect of Balcony Thermal Breaks on Building Thermal and Energy Performance: Field Experiments and Energy Simulations in Chicago, IL

**Irina Susorova [1,2,*], Brent Stephens [1,*]** 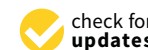 **and Benjamin Skelton [2]**

[1]  Department of Civil, Architectural, and Environmental Engineering, Illinois Institute of Technology, Chicago, IL 60616, USA

[2]  Cyclone Energy Group, Chicago, IL 60605, USA

*  Correspondence: irinasusorova@gmail.com (I.S.); brent@iit.edu (B.S.); Tel.: +1-312-882-0632 (I.S.)

**Abstract:** A common envelope performance problem in buildings is thermal bridging through balcony slab connections, which can be improved with the use of commercially available thermal break products. Several prior studies have used simulation-based and/or hot box test apparatus approaches to quantify the likely effect of balcony thermal breaks on effective thermal resistance of building enclosures. However, in-situ measurements of thermal performance in real buildings remain limited to date. This study uses a combination of field measurements and models to investigate the effects of installing balcony thermal breaks on the interior surface temperatures, effective thermal resistance, and annual building energy consumption. For the field experiment, yearlong measurements were conducted on the 13th floor of a 14-story multi-family building in Chicago, IL, in which thermocouple sensors were embedded into eight balconies and their adjacent interior floor slabs just before concrete was poured to complete the construction. The eight balconies included four control balconies without thermal breaks and four thermally-broken balconies with a commercially available thermal break product installed. The experimental data were then combined with 2-D heat transfer modeling and whole building energy simulations to investigate the impacts of the thermal break product installation on the envelope thermal resistance and overall energy use in the case study building as well as in several more generic building designs with simpler geometries. The results demonstrate that although the balcony thermal breaks helped regulate interior slab temperatures and improved the effective thermal resistance of the curtain wall enclosure assembly by an estimated ~14% in the case study building, the predicted effect on annual energy consumption in all modeled building types was small (i.e., less than 2%). The results also highlight the importance of paying careful attention to envelope design details when using thermal break products and considering the use of thermal break products in combination with other energy efficiency strategies to achieve high performance enclosures.

**Keywords:** Balcony thermal breaks; thermal bridges; building envelopes; energy analysis; THERM; multi-family residential buildings

## 1. Introduction

One of the most prevalent envelope performance problems in buildings is thermal bridging. A very common thermal bridge in the building envelope of mid- and high-rise residential buildings occurs at the balcony slab connection [1,2]. These balcony slab connections are commonly not thermally broken, which can lead to poor building thermal performance and increased energy consumption due to increased heat loss in winter and heat gain in summer [3–6]. Another problem with balcony slab connections is that cold indoor floor slab surfaces at the building perimeter can contribute to occupant thermal discomfort in winter.

One of the most widely recommended solutions to the balcony slab thermal bridge problem is to introduce thermal breaks [1,2,4,7–9] or insulated concrete curbs [10]. Thermal breaks are thermally insulating elements embedded in the structure that separate the balcony from the floor slab and reduce heat transfer through the connection. Thermal breaks are composed of thermal insulation between concrete slabs and structural reinforcement that is connected to the reinforcement bars in the floor and balcony slabs. The insulation materials typically include extruded polystyrene, expanded polystyrene, or mineral wool, all of which have similar levels of thermal conductivity (i.e., 0.025-0.040 W/m·K) [8]. Structural reinforcement can be made of regular rebar steel or stainless steel that is less thermally conductive. Thermal breaks do not completely stop heat transfer through the balcony connection (i.e., high rates of heat transfer can still occur through the metal reinforcement, which can occupy 10%–20% of the cross-sectional area [4]), but they can substantially reduce heat transfer through the remaining 80%–90% of the concrete slab area. The thermal break performance is affected by the reinforcement bar diameter, the number of structural elements, the exterior wall U-value, and the size of balcony slabs. Some research has also been conducted to date on increasing thermal break performance through the use of aramid fiber [9] and fiber-reinforced polymer [11] structural reinforcement that is less thermally conductive than stainless steel reinforcement.

While using balcony thermal breaks is a common energy-efficiency strategy in Europe and Canada, it is relatively new in the United States market. Numerous studies of the effects of balcony thermal breaks have been conducted using simulation-based approaches, while a smaller number have utilized experimental measurements under controlled conditions, for example, using a hot-box apparatus.

When using a simulation approach, it is important to model thermal bridges using the most accurate methodology. The two commonly used methods of modeling thermal bridges include (a) the equivalent U-value method (i.e., where thermal bridges are modeled as part of a weighted average U-value for the entire envelope) and (b) the direct three-dimensional modeling method (i.e., where thermal bridges are explicitly drawn on envelope surfaces with actual dimensions and material properties). However, these methods do not predict building energy performance with thermal bridges equally well. For example, a study by Ge et al. [12] found that it is best to use the three-dimensional heat transfer method when predicting the impact of balcony slab thermal bridges in multifamily high-rise buildings with concrete structures.

Many researchers have also evaluated the effect of adding balcony thermal breaks on the overall building energy consumption. A simulation-based study by Ge et al. evaluated thermal improvements to various balcony connection details and their impact on the total building energy performance for case studies with and without balcony thermal breaks in different Canadian climates. Ge et al. found that the inclusion of thermal breaks in balcony connections can potentially reduce annual heating energy consumption by 5–11% [4]. A follow-up simulation-based study by Baba et al. for the same climate zone found that the inclusion of thermal breaks in balcony connections can potentially reduce annual heating energy consumption by 7–8% but increase annual cooling consumption by 4–12%, and that the effect will vary depending on the climate, window area, and adjacent wall types [13]. Similarly, a study by Hardock et al. found that balcony thermal breaks could reduce annual building energy use by 7.3% in the Chicago climate [5].

Reducing thermal bridges through balcony connections also helps improve indoor thermal comfort for building occupants. A simulation-based study by Finch et al. found that indoor balcony slab temperatures can be increased by 4 °C–7 °C when balcony thermal breaks are included in buildings located in cold climates [6]. Another study by Dikarev et al. evaluated balcony connections with thermal breaks in a hot-box apparatus and found that the inclusion of balcony thermal breaks can help increase the indoor slab temperature by up to 8 °C [14]. The alternative solution to the balcony thermal bridge problem, insulated concrete curbs, can also improve indoor slab conditions by raising its temperature by approximately 4 °C [10].

Considering the aforementioned beneficial effects of balcony thermal bridges, it is important not to overestimate the energy and cost savings associated with reducing thermal bridges, including the

installation of balcony thermal breaks. It is equally necessary to consider the financial feasibility of the balcony thermal break solution as reported by Evola et al. [15]. This study, which evaluated the economic feasibility of correcting envelope thermal bridges in mild climates, found that although the elimination of the envelope thermal bridges is an effective measure to reduce heating energy usage, it is not always economically feasible because of long payback periods (18–20 years).

Despite the existing body of research to date, there is very little measured quantitative information available on how thermal breaks can affect in-situ building thermal performance and/or overall energy consumption. This study seeks to fill this information gap by evaluating the effects of balcony thermal breaks using (i) field measurements of balcony slab and surface temperatures in an actual constructed building in the United States and (ii) 2D thermal modeling and whole building energy simulations in the same case study building, as well as in several more generic building designs with simpler geometries.

## 2. Materials and Methods

### 2.1. Experimental Methods

To experimentally characterize the in-situ thermal performance of balcony thermal breaks, a 14-story multi-family residential building located in Chicago, IL, which was under construction at the start of the study, was recruited for field measurements. The building had a total of 110 south-facing balconies, all of which except four balconies on the 13th floor, were constructed following the conventional practice of monolithic floor slabs extending beyond the vertical wall plane to create balconies without thermal breaks. The other four balconies on the 13th floor had commercially available thermal break products installed to separate the floor slab and balcony (Schock Isokorb® CM40 product) (Figure 1). Note that this was not the best performing thermal break product available from the manufacturer (Schock Isokorb® CM30, CM20, and CM10 products have higher nominal insulation values for the same slab thickness) [16]. The installation of the thermal break products on the site is shown in Figure 2.

The thermal break product introduced an interruption in the floor-to-balcony slab using 80 mm thick thermal insulation with stainless steel structural reinforcement (part of thermal break product) protruding through the insulation and connecting with the building rebar on either side. The area of structural reinforcement embedded in the thermal break product was estimated to be ~1.7% of the product's total cross-sectional area based on the number and type of rebar used: 32 12 mm- and 16 6 mm-rebars for the short balcony side and 80 12 mm- and 20 8 mm-rebars for the long balcony side, according to the building's structural details.

A total of eight balconies on the 13th floor were selected for making long-term in-situ temperature measurements inside the concrete on both the slab and balcony sides, including: The four thermally broken balconies and four conventional balconies that were selected on the same floor with the same south-facing orientation and with similar geometries to those balconies that were thermally broken. At this elevation, there were no obstructions or shading that could have influenced the results. A typical floor plan with the two balcony groups (i.e., control balconies and thermally-broken balconies) is shown in Figure 3. The typical balcony dimensions were 4.3 m by 2.3 m and the building envelope at the south façade consisted of floor-to-ceiling insulated glazing units.

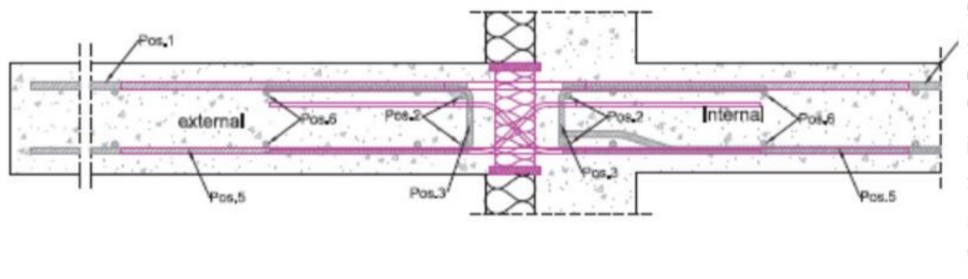

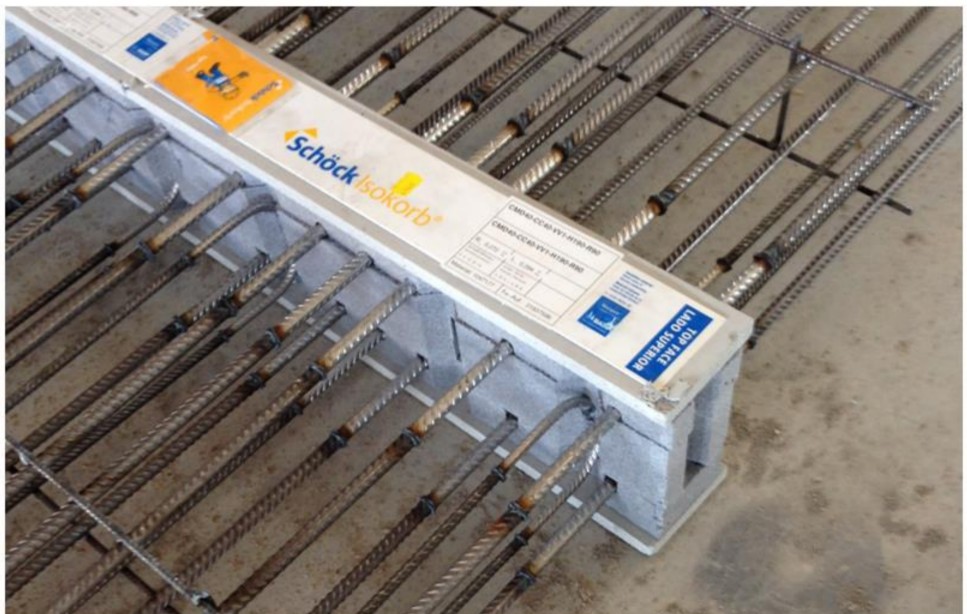

**Figure 1.** Cross section and detail of the installed balcony thermal break.

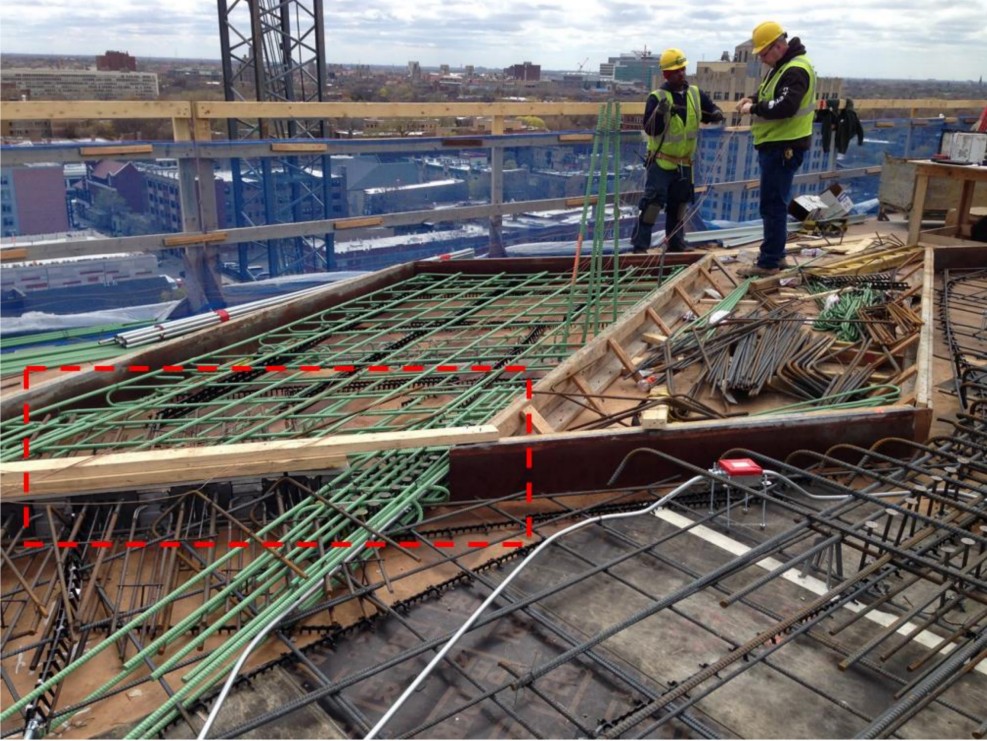

**Figure 2.** Balcony thermal break product and on-site installation.

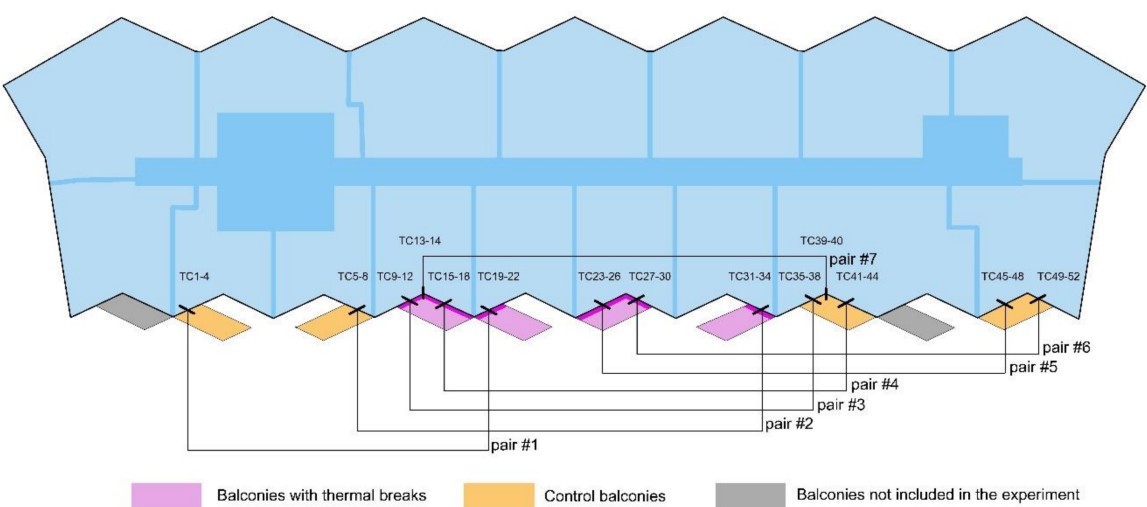

**Figure 3.** Floor plan showing the eight studied balconies on the 13th floor: Four balconies with thermal breaks and four conventional balconies without thermal breaks. Each balcony comparison pair was chosen based on similar geometries. Comparison balcony pairs are labeled pair #1-7. TC = thermocouple.

A total of 52 type-K thermocouple (TC) sensors were embedded into the eight selected balconies and their adjacent interior floor slabs just before concrete was poured to complete construction of the floor slabs and balconies (Figure 4).

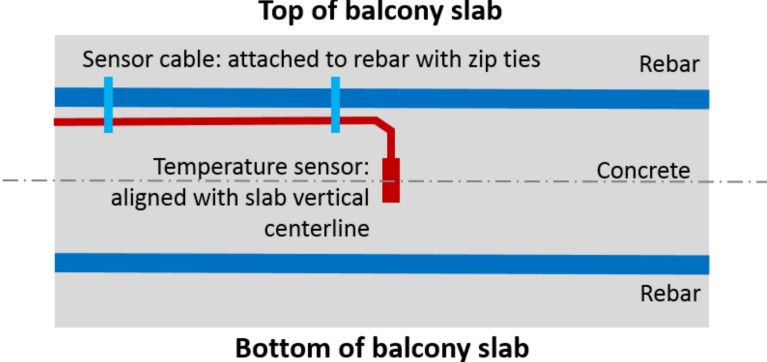

**Figure 4.** Cross section through a typical balcony.

Each thermocouple was aligned vertically with the balcony centerline and attached to the structural rebar with wire to remain reasonably well fixed during the concrete pour. Each balcony and corresponding housing unit had one exterior thermocouple installed approximately 0.6 m from the connection between the floor-to-ceiling glazing units and the slab, and three interior thermocouples installed at 0.3 m, 0.6 m, and 0.9 m from the wall-slab connection in the opposite linear direction (i.e., toward the inside of the housing unit). Two of the balconies that had thermal break products installed on both of their two sides also had two additional thermocouples embedded in the exterior slab between the thermal breaks and in the interior floor slab at 0.3 m from the perimeter. Since the building balconies were rhomboid in shape and all were oriented at slightly different angles, the thermocouple readings were grouped into seven pairs. Each pair was made to combine a control balcony and a corresponding thermally-broken balcony of the same geometric configuration. All thermocouple wires were run through the interior slab back to a common corridor where they were connected to Onset HOBO UX120 4-channel thermocouple loggers that remained externally accessible to the research team to periodically download the logged data. A detailed drawing of one of the equipment installations for a single balcony (pairs #5 and #6) is shown in Figure 5.

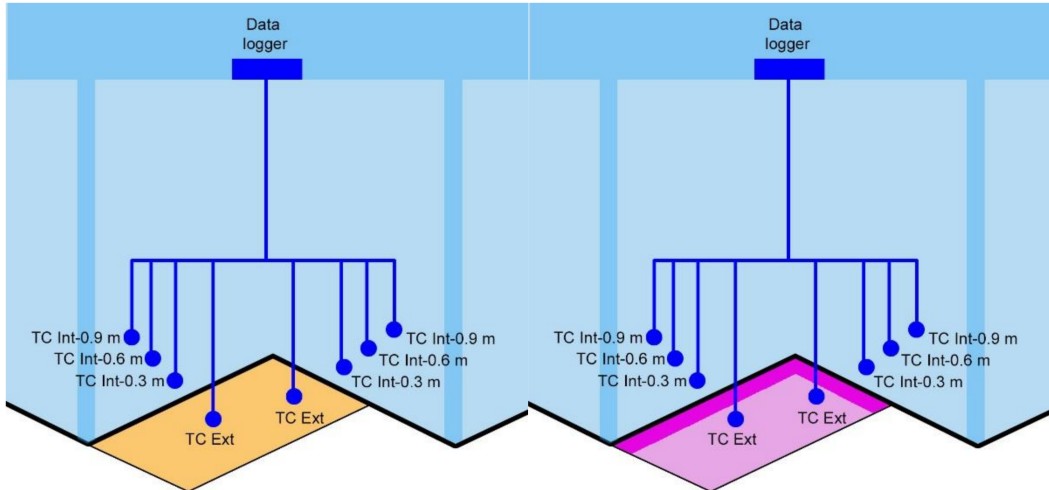

**Figure 5.** Detail of the experimental setup for balcony pair #5 and #6. The conventional balcony is shown on the left and the thermally-broken balcony is shown on the right (this particular balcony is thermally-broken in two places). TC = thermocouple; Int = interior; Ext = exterior.

In addition to slab temperatures, the research team also measured local weather conditions using an Onset HOBO U30 weather station installed on the roof of the building. The floor slab, balcony temperatures and weather data were collected at five-minute intervals between February 2016 and January 2017. The interior and exterior temperature data were used to compare the effect of the balcony thermal breaks on indoor slab temperatures with their conventional balcony counterparts. Then, the collected measurements were used to estimate the effective thermal resistance of the control and thermally-broken balcony connections, which were also compared to the effective thermal resistance values obtained using a two-dimensional heat transfer analysis using THERM. The 2-D thermal analysis was also used to investigate the impact of various enclosure configurations and assumptions for physical modeling domains on the resulting effective thermal resistances of both conventional balconies and balconies with a thermal break.

### 2.2. Estimating the Effect of the Thermal Breaks on Envelope Thermal Resistance

#### 2.2.1. Using Experimental Data

The measured slab temperature data were first used to estimate the effective increase in thermal resistance achieved by the thermal break product, again using only data from balcony pair #1. The instantaneous effective thermal resistance estimates for both the control and thermal break balconies were approximated using the 1-D steady state equation for thermal conduction through a material (Equations (1) and (2)).

$$Q_{control} = \frac{\Delta T_{control}}{R_{control}} \tag{1}$$

$$Q_{thermal\ break} = \frac{\Delta T_{thermal\ break}}{R_{thermal\ break}} \tag{2}$$

where $Q$ is the heat flux through the material (W/m$^2$), $\Delta T$ is the temperature differential between the exterior and interior environment (K or °C), and $R$ is the effective thermal resistance of the assembly (K·m$^2$/W).

It was assumed that conductive heat transfer occurs from the interior towards the exterior perimeter of the slab in winter time and that there is no storage at the temperature sensor nodes in the long term. It was further assumed that the 1-D conductive heat flux ($Q$) between the interior sensors located 0.9 m and 0.3 m away from the perimeter (TC Int 0.9 m and TC Int 0.3 m) was equal to the 1-D heat flux between the interior sensor located 0.3 m away from the perimeter and the exterior sensor

(TC Int 0.3 m and TC Ext). Thus, the effective 1-D heat flux through both the interior floor slabs and slab-to-balcony connections could be approximated using Equation (3). Note that this is a necessary oversimplification because there are limited other options available for analyzing the field data.

$$Q = \frac{(TC\ Int_{0.9\ m} - TC\ Int_{0.3\ m}) \cdot k_{slab}}{\Delta x_{slab}} = \frac{(TC\ Int_{0.3\ m} - TC\ Ext)}{R_{in-out}} \tag{3}$$

where $k_{slab}$ is the thermal conductivity of the reinforced concrete floor slab (W/m·K), $\Delta x_{slab}$ is the distance between the two sensors in the floor slab (0.6 m), and $R_{in-out}$ is the effective 1-D thermal resistance of the material as measured between 0.3 m toward the interior and the exterior measurement location. In other words, the effective 1-D thermal resistance calculated here accounts for a combination of: Approximately 0.3 m of interior slab and concrete; the thermal break product itself; and approximately 0.6 m of exterior slab and concrete. Equation (4) is then used to calculate the effective 1-D thermal resistance for this entire length of both the control and thermally-broken floor slabs. The thermal conductivity of the reinforced concrete slab ($k_{slab}$) was estimated to be 2.52 W/m·K using a cross-sectional area-weighted combination of 1.8 W/m·K for concrete (occupying 98.5% of the floor slab volume) and 50 W/m·K for steel rebar (occupying 1.5% of the floor slab cross-sectional area) [16].

$$R_{in-out} = \frac{(TC\ Int_{0.3\ m} - TC\ Ext) \cdot \Delta x_{slab}}{(TC\ Int_{0.9\ m} - TC\ Int_{0.3\ m}) \cdot k_{slab}} \tag{4}$$

### 2.2.2. THERM Modeling: Case Study Building

Next, a series of steady-state 2-D heat transfer simulations of the case study balconies were conducted using THERM in an attempt to confirm the reasonableness of the estimates of 1-D thermal resistance made using the experimental data. There were two types of balcony connections among all balcony configurations: Along the long side (4.3 m) and along the short side (2.3 m). Each balcony slab connection consisted of cast-in-place concrete and structural reinforcement steel. The thermal breaks did not provide complete thermal separation of the balcony slabs and interior floor slabs. Due to their complex shapes, the balcony slabs had to be supported at the corners with poured-through concrete and regular reinforcement. The THERM analysis included vertical sections through both connections, the sections extending 0.3 m inside from the building perimeter (matching the positions of the thermocouple sensors in the field study), and 0.2 m of the exterior wall above and below the floor slab (Figure 6). Using this geometry, the thermal effects of adjacent envelope elements could also be investigated in addition to modeling the balcony connections alone (which has been done previously in the literature on the effectiveness of thermal breaks).

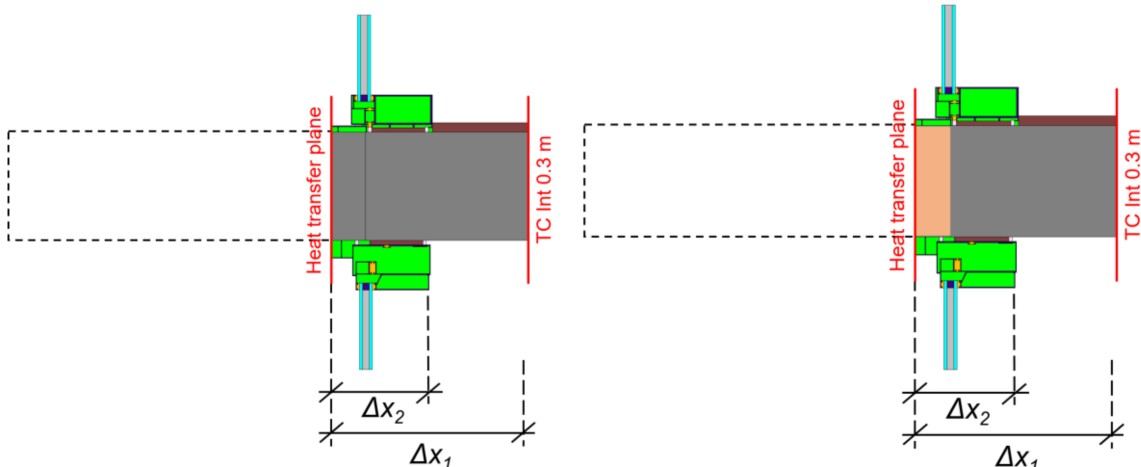

**Figure 6.** Vertical sections through the control (left) and thermally-broken (right) balcony connections.

　　　The envelope material thermal properties used in the THERM analyses were based on widely available material properties tables, including those reported in the American Society of Heating, Refrigerating, and Air-Conditioning Engineers (ASHRAE) Handbook of Fundamentals [17] and the thermal break product manufacturer's literature for an 80 mm thick product [16] (Table 1). The reinforced concrete thermal conductivity was estimated to be ~2.52 W/m·K as described in Section 2.2.1. The conductivity of the thermal break product was 0.294 W/m·K as provided by the manufacturer. The THERM analysis was conducted for steady-state conditions assuming an outdoor temperature of 0 °C and an indoor temperature of 21 °C and air film coefficients of 35 W/m²·K for the exterior wall surface, 9.3 W/m²·K for the interior ceiling surface, and 6.1 W/m²·K for the interior floor surface.

**Table 1.** Material thermal properties used in the THERM analysis.

| Thermal Conductivity | (W/m·K) | Source |
|---|---|---|
| Concrete | 1.8 | [16] |
| Reinforcement steel | 50 | [16] |
| Stainless steel | 15 | [16] |
| Th. Break insulation (polystyrene hard foam) | 0.031 | [16] |
| Aluminum (anodized) | 237 | [17] |
| Gypsum board | 0.159 | [17] |
| Hardboard (medium density) | 0.105 | [17] |
| Face brick | 1.311 | [17] |
| Cavity insulation | 0.06 | [17] |
| Continuous insulation | 0.029 | [17] |
| *Composite materials* | | |
| Reinforced concrete | 2.523 | Calculated |
| Thermal break | 0.294 | [16] |

　　　Following the methodology described in [2], the heat transfer plane was set along the exterior wall surface. According to this method, the elements extending past the building envelope (i.e., balcony slabs) are not important for heat transfer calculation.

*2.3. Energy Simulation Methods*

　　　Once the effective thermal resistance of the measured balcony thermal breaks was estimated based on the combination of experimental and 2-D thermal analysis results, it was then possible to evaluate the likely effects of installing thermal breaks in all balconies on the overall building energy performance. This was done using two simulation case studies. First, the case study building that was used for field measurements was simulated assuming all balconies were thermally broken, and the results were compared to the building without any thermally broken balconies. Second, because the unique geometry of the case study building makes the results difficult to generalize, energy simulations were also conducted with and without thermal breaks on balconies of multi-family residential buildings with simpler geometries that are more representative of typical buildings. The latter approach also allowed for a deeper parametric analysis of several important building characteristics that were assumed to be likely to influence the results.

2.3.1. Energy Simulations with the Case Study Building

　　　An energy model of the case study building from the field measurements was created in IES<VE>2017, and a set of dynamic annual energy simulations were performed for the climate of Chicago using the O'Hare International Airport typical meteorological year (TMY3) weather file. The building energy model included 99 south-facing residential units with balconies with a total floor area of 8,332 m². The north-facing units without balconies were excluded from the simulations for simplicity. The energy simulations were conducted for the cases (a) without balcony thermal breaks and (b) with thermal breaks using the effective thermal resistance calculated from the field measurement data. For

the balcony thermal break scenarios, it was assumed that all balconies in the building had thermal breaks rather than just the four that were measured in the field experiments. The annual energy simulations included all energy end use categories associated with a multi-family residential building, such as space heating, space cooling, heat rejection, fans, pumps, interior lighting, receptacle loads, elevators, and domestic hot water.

2.3.2. Parametric Energy Simulations

The building used in the experimental monitoring had a unique design, which made difficult to make broader generalizations about balcony thermal break energy performance. Therefore, in order to evaluate the energy savings potential of balcony thermal breaks for a variety of more common multi-family residential building designs, parametric whole-building energy simulations were conducted using the energy models of more generic multi-family residential buildings with simpler geometries. The range of input parameters for the parametric study included:

- Number of stories: 5 and 20
- Window to wall ratio (WWR): 40% and 100%
- Balcony geometry: Single and continuous.

The number of stories was selected to represent a typical mid-rise building (5 stories) and a typical high-rise building (20 stories). The WWR of 40% represents the recommended value by the commonly used building energy codes of ASHRAE Standard 90.1 [18]. The WWR of 100% represents an inefficient but very common architectural practice in north American multi-family residential buildings. Each residential unit had either two 6.0 m x 1.8 m balconies for both orientations (for single balcony geometry) or a continuous 1.8 m deep balcony wrapped around the unit perimeter (for continuous balcony geometry). In the case of continuous balconies, the balcony connections were assumed to occupy 5.0% (5-story) and 4.9% (20-story) of the total building exterior wall area; in the case of single balconies, they were assumed to occupy 2.2% (5-story) and 1.1% (20-story).

Other modeling assumptions are listed below and shown in Figures 7 and 8:

- The modeled building was square in the plan 27 m by 27 m with four residential units per floor occupying the southeast, southwest, northwest, and northeast corners. The square floor plan was selected to minimize the effects of building orientation. The floor-to-floor height was 3 m. Each three-bedroom unit was approximately 162 m$^2$ with an assumed occupancy of four people per unit. Thus, the five-story building had 20 residential units with a total area of 3,660 m$^2$ and the 20-story building had 80 units with a total area of 14,680 m$^2$.
- The residential floors were placed above grade and had no thermal adjacency to the ground.
- The building envelope assemblies included the roof with RSI 3.52 K·m$^2$/W insulation and the exterior brick veneer metal stud walls with RSI 2.34 K·m$^2$/W cavity and RSI 1.76 K·m$^2$/W continuous insulation. The building fenestration consisted of insulated glazing units with U-value of 2.38 W/K·m$^2$ (U-0.42) and solar heat gain coefficient of 0.4.
- The building internal heat gains (lighting, receptacle, and people loads) were based on the default values of ASHRAE 90.1 Standard [18].
- The residential unit level heating and cooling was provided by the hot-water loop served by two natural-draft boilers (80% efficiency) and single-zone packaged air-conditioning units (10.2 EER).
- Outdoor air ventilation was provided through common corridors by the central makeup air unit (DX cooling with energy efficiency ratio (EER) of 11 and 80% efficient furnace) with a variable frequency drive.
- The domestic hot water flow was assumed to be 26.5 L/h per residential unit served by a dedicated 80% efficient natural gas heater.

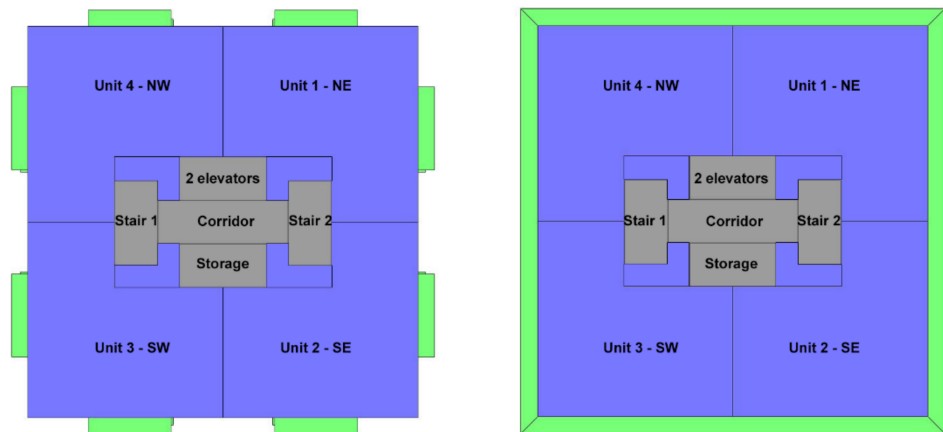

**Figure 7.** Typical building floor plans with single balconies (left) and continuous balconies (right).

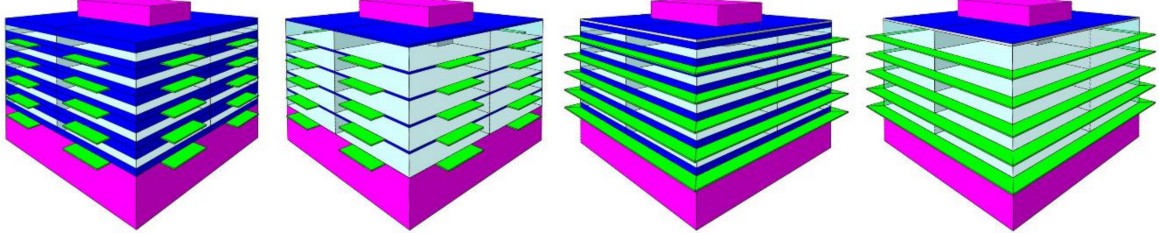

**Figure 8.** Five-story building models with different window to wall ratio (WWR) and balcony geometry.

## 3. Results and Discussion

### 3.1. Measured Effects of the Thermal Breaks on Interior and Exterior Surface Temperatures

The measured indoor and outdoor slab temperatures within all balcony pairs showed high variability due to the different balcony geometries, orientation angles, and solar exposure (see Figures A1–A6 in the Appendix A for the full data set). Additionally, an exterior shear wall connected to the balconies at different locations and different angles shaded the floor slabs differently during the day, which added complexity to the interpretation of the collected data. Moreover, some of the balconies were measured during unoccupied periods (i.e., before residents moved in), while others were measured during occupied periods after residents moved in. The occupied residences were typically conditioned to higher thermal comfort standards (i.e., average of 21.0 °C in heating and 24.0 °C in cooling seasons), while the unoccupied residences were not (i.e., average of 15.5 °C in heating and 26.7 °C in cooling seasons). Therefore, it was difficult to control for these myriad factors on all balcony pairs. In fact, only balcony pair #1 had similar indoor thermal conditions for both the control and thermally-broken balconies during the entire experiment (both were occupied) and therefore, it was selected as the pair that was most appropriate for comparing the effects of the thermal break on the interior slab temperature measurements for all subsequent calculations (Table 2).

The average exterior slab temperatures were similar between the control and thermally broken balconies in balcony pair #1 for both time frames (i.e., ~4 °C during winter and ~26 °C during summer). As expected, the interior slab temperatures at both 0.9 m and 0.3 m from the thermally-broken balcony for balcony pair #1 were on average 1.2–1.6 °C lower than those of the control unit during the summer periods (June-August) and 1.1–1.3 °C higher in the winter periods (November-January). Thus, the introduction of a balcony thermal break clearly led to some improvement in regulating the interior slab temperatures in this balcony pair and thus, would be expected to potentially improve indoor thermal comfort by regulating floor temperatures during both seasons.

Figure 9 shows the temperature readings for balcony pair #1 for the three winter months of the monitoring periods (November, December, and January) when the effect of thermal breaks was the most pronounced. During periods of very cold ambient temperatures (i.e., below 0 °C), the interior

slab temperatures adjacent to the thermally broken balcony were as much as 4 °C warmer than those of the control unit, which further illustrates a positive effect that the thermal break products had in regulating interior slab temperatures. Notably, these measurements are very close to the experimental findings by Finch et al. [6], Murad, et al. [10], and Dikarev et al. [14].

**Table 2.** Slab temperature of the control and thermally-broken balconies (pair #1).

| | Int. Slab Temperature 0.9 m from Perimeter (°C) | | Int. Slab Temperature 0.3 m from Perimeter (°C) | | Ext. Slab Temperature (°C) | |
|---|---|---|---|---|---|---|
| | CT (TC01) | TB (TC19) | CT (TC03) | TB (TC21) | CT (TC04) | TB (TC22) |
| *June-August* | | | | | | |
| Max | 28.5 | 26.6 | 29.7 | 26.9 | 36.5 | 35.4 |
| Average | 26.3 | 25.0 | 26.3 | 24.7 | 26.3 | 25.8 |
| Min | 22.1 | 23.1 | 21.1 | 21.6 | 15.2 | 15.8 |
| St. Dev | 1.1 | 0.6 | 1.6 | 0.9 | 3.9 | 3.4 |
| Sample size | 26,496 | 26,496 | 26,496 | 26,496 | 26,496 | 24,936 |
| *November-January* | | | | | | |
| Max | 26.5 | 25.6 | 25.5 | 24.3 | 26.7 | 26.0 |
| Average | 21.6 | 22.8 | 17.4 | 18.7 | 3.9 | 4.0 |
| Min | 16.6 | 19.0 | 12.4 | 15.0 | -13.9 | -14.5 |
| St. Dev | 1.9 | 1.0 | 3.1 | 2.0 | 7.9 | 7.7 |
| Sample size | 26,496 | 26,496 | 26,496 | 26,496 | 26,496 | 26,496 |

CT - control balcony, TB - thermally-broken balcony, TC – thermocouple.

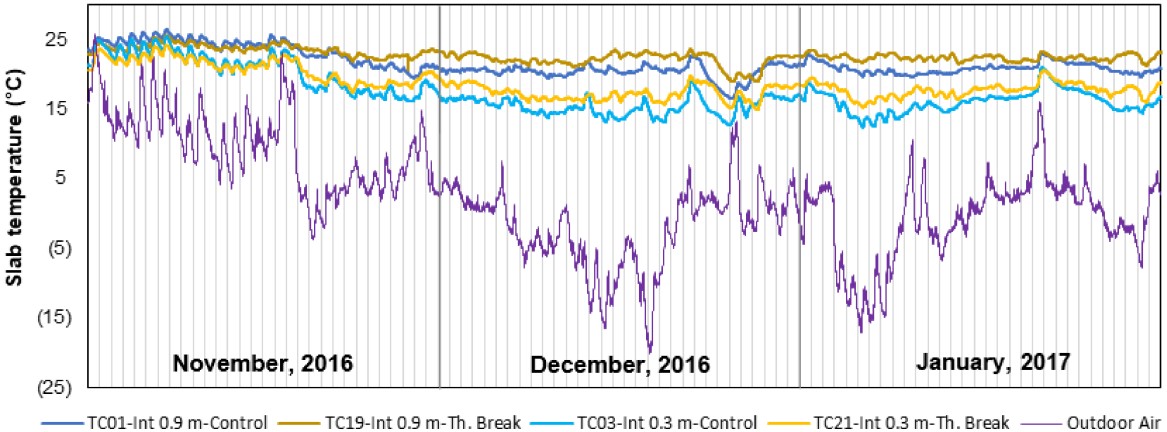

**Figure 9.** Interior floor slab temperature of the control and thermally-broken balconies (pair #1).

*3.2. The Effect of the Thermal Breaks on Envelope Thermal Resistance*

3.2.1. Experimental Results

In practice, the thermal resistance of insulation materials is rated under standardized test conditions in a controlled laboratory environment, typically with a mean simulated indoor air temperature of 21 °C, 50% relative humidity, no air movement, no exposure to solar radiation, and with a relatively high indoor/outdoor temperature differential of at least 20 °C [19]. In order to provide an approximation of the effective 1-D thermal resistance of the balcony assemblies, extending the measurement length from ~0.3 m to the interior to ~0.6 m to the exterior, that is comparable to that measured at standardized rating conditions, our analysis (described in Section 2.2.1) used only the data collected from periods when the indoor/outdoor temperature differential was greater than 20 °C using Equation (4) and when no solar radiation was present, as shown in Figure 10.

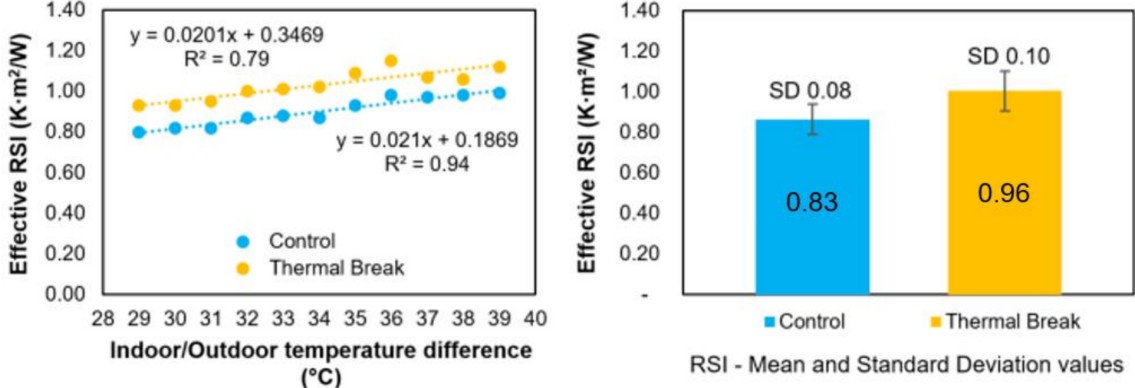

**Figure 10.** The effective 1-D thermal resistance for approximately 1 m length from ~0.3 m to the interior of the curtain wall to ~0.6 m toward the exterior of the balcony for balcony pair#1 estimated using data from November to January during periods when the exterior–interior $\Delta T$ was greater than 20 °C and when no solar radiation was present.

Using this method, the average effective 1-D thermal resistance of the approximately 1 m of floor slab sections was estimated to be ~0.83 K·m$^2$/W (±0.08 K·m$^2$/W) for the control balcony and ~0.96 K·m$^2$/W (±0.10 K·m$^2$/W) for the thermally-broken balcony, averaged over all of the winter periods when the indoor/outdoor temperature differential was greater than 20 °C. The effective resistance of the thermal break product in this installation can be calculated as the difference between the resistances of the control and thermally broken slabs (as suggested by [2]), which results in an average value of 0.12 K·m$^2$/W. These estimates demonstrate that in this building, the effective 1-D thermal resistance of the balcony with the thermal break product installed, as measured across a length of approximately 1 m, was increased by an average of approximately 14% compared to the monolithic control slab balcony. It is worth noting that if a better performing insulated balcony thermal break product by the same manufacturer had been installed in the studied building, the increase in the effective thermal resistance of this connection likely would have been higher.

3.2.2. THERM Modeling Results: Case Study Building

Figure 11 shows the modeled temperature distributions across the studied sections predicted using THERM. The resulting modeled effective 2-D thermal resistance of the existing balcony slabs (spanning from ~0.3 m to the interior to the exterior wall surface) was estimated to be RSI 0.23 K·m$^2$/W for the control balcony and RSI 0.52 K·m$^2$/W for the thermally-broken balcony. Thus, the effective resistance of the thermal break product was estimated to be 0.29 K·m$^2$/W. The value of the thermal break RSI calculated with THERM (0.29 K·m$^2$/W) is higher than that estimated from the field measurements (0.12 K·m$^2$/W). The differences in the results obtained by the THERM analysis and calculated from the measured data may be explained in part by the following factors:

- The estimates from the measurements considered the length of the exterior balcony slab (0.6 m) while the THERM analysis did not.
- The actual weather conditions experienced by the building (e.g., highly varying wind speed/direction and interior-exterior $\Delta T$) were different from the ideal and static conditions used in the THERM analysis.
- The THERM analysis does not capture 3-D heat transfer effects from the adjacent balconies and shear walls of the case study building.

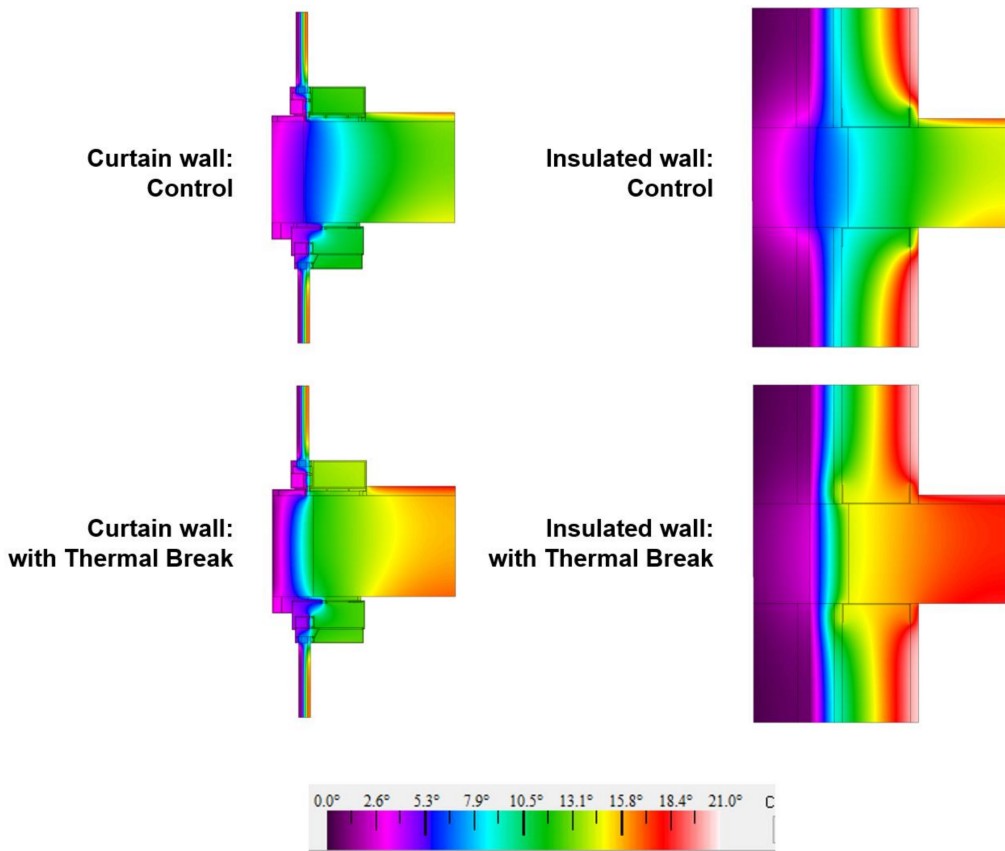

**Figure 11.** Color gradient of temperature distributions for the different balcony connection options and the installed thermal break product.

Despite these differences, the combination of the measured and modeled results demonstrate that the thermal break product installed in the studied building performs approximately as intended, with its in-situ measured effective 1-D thermal resistance being relatively close to the effective 1-D thermal resistance predicted using a 2-D model (i.e., within 0.17 K·m$^2$/W). Moreover, these results also demonstrate that the THERM modeling approach can be useful for modeling thermal breaks in other balcony and wall geometries, especially when accounting for rebar penetrating through the break products.

### 3.2.3. THERM Modeling: Alternative Configurations of the Case Study Building

It is worth noting that the measured balcony connections were somewhat unique and not particularly thermally efficient because the envelope assembly separating the outdoor and indoor environments was a double-glazed aluminum-frame curtain wall that abutted the floor slab above and below. To evaluate whether the adjacent envelope assembly elements would affect the balcony connection thermal performance, an additional analysis was conducted in THERM for a common exterior wall assembly type with the same thermal break product: A 0.14 m metal-stud wall with RSI 2.34 K·m$^2$/W cavity and RSI 1.76 K·m$^2$/W continuous insulation and with brick veneer cladding. This insulated exterior wall assembly was based on the minimum energy code envelope performance requirements for Climate Zone 5 (Chicago) [18]. The resulting effective thermal resistances for all modeled options are shown in Table 3, which includes the thermal resistance values calculated for the two depths of the exterior wall vertical section: ($\Delta x_1$) corresponding to the approximately 0.3 m distance between the thermal sensors TC Int 0.3 m and the exterior wall surface and ($\Delta x_2$) corresponding to the depth of the exterior walls only (Figure 6). The latter is the RSI value that is commonly used in envelope thermal calculations and energy simulations.

**Table 3.** Floor slab thermal properties calculated in THERM with two assumed wall types (CM40 thermal break product).

| Exterior Wall Type | Effective Thermal Resistance (K·m²/W) | | | | | |
| | Control Balcony | | Thermally-Broken Balcony | | Balcony Thermal Break | |
| | $\Delta x_1$ * | $\Delta x_2$ ** | $\Delta x_1$ * | $\Delta x_2$ ** | $\Delta x_1$ * | $\Delta x_2$ ** |
| Curtain wall | 0.23 | 0.34 | 0.52 | 0.68 | 0.29 | 0.34 |
| Insulated wall | 0.36 | 0.75 | 0.73 | 1.11 | 0.37 | 0.36 |

* Between the location of sensors TC Int 0.3 m and exterior wall; ** Only the thickness of exterior wall (0.19 m for curtain wall and 0.34 m for insulated wall).

These results show that the type of exterior wall adjacent to the floor slab can influence the effect that the balcony thermal break can have on thermal performance. In the case of the curtain wall, thermal breaks increased the exterior wall thermal resistance by an additional RSI 0.29 K·m²/W (nominal 0.34 K·m²/W) while in the well-insulated wall case, thermal breaks increased the exterior wall thermal resistance by an additional RSI 0.37 K·m²/W (nominal 0.36 K·m²/W).

As previously mentioned, the same manufacturer also offers better-performing balcony thermal break products for 180 mm slab thickness than the product selected for the installation (CM40 with thermal conductivity of 0.294 W/m·K). The additional THERM analyses (Table 4) show the calculated RSI results for the balcony connections assuming that the better-performing thermal break product had been installed (i.e., C10 with thermal conductivity of 0.190 W/m·K) [16]. With the better product, thermal breaks increased the exterior wall thermal resistance by an additional RSI 0.44 K·m²/W (nominal 0.51 K·m²/W) for the curtain wall and by an additional RSI 0.55 K·m²/W (nominal 0.56 K·m²/W) for the well-insulated wall.

**Table 4.** Floor slab thermal properties calculated in THERM with two assumed wall types (CM10 thermal break product).

| Exterior Wall Type | Effective Thermal Resistance (K·m²/W) | | | | | |
| | Control Balcony | | Thermally-Broken Balcony | | Balcony Thermal Break | |
| | $\Delta x_1$ * | $\Delta x_2$ ** | $\Delta x_1$ * | $\Delta x_2$ ** | $\Delta x_1$ * | $\Delta x_2$ ** |
| Curtain wall | 0.23 | 0.34 | 0.66 | 0.85 | 0.44 | 0.51 |
| Insulated wall | 0.36 | 0.75 | 0.91 | 1.31 | 0.55 | 0.56 |

* Between the location of sensors TC Int 0.3 m and exterior wall; ** Only the thickness of exterior wall (0.19 m for curtain wall and 0.34 m for insulated wall).

The closest study to which these results could be compared is the extensive thermal bridging study by Morrison Hershfield [2], which calculated the thermal resistance of balcony thermal breaks in a curtain wall installation (Detail 8.1.7 in their report). In that study, the RSI of the control balcony slab was calculated as 0.20 K·m²/W and the RSI of the thermally-broken balcony slab was calculated as 0.83 K·m²/W. Thus, the thermal break RSI could be estimated by adding an additional RSI of 0.63 K·m²/W, which is slightly higher than what was found through both the measurements and THERM modeling in the case study building with the Schock Isokorb® CM40 product installed (0.29–0.37 K·m²/W), but is consistent with THERM modeling with the better performing Schock Isokorb® CM10 product.

*3.3. Energy Simulation Results*

Next, the resulting RSI values from Table 3 corresponding to the depth of the exterior walls only ($\Delta x_2$) were used in a series of whole building energy simulations described in Section 2.2. The analyses were conducted for the two sets of balcony connection RSI values: (a) RSI for the curtain wall and (b) RSI for the insulated opaque wall.

3.3.1. Case Study Building Results

An energy analysis was first conducted to evaluate the potential improvement in energy performance of the case study building. The RSI values used in this analysis were 0.34 K·m$^2$/W for control and 0.68 K·m$^2$/W for thermally-broken balconies (Table 5). The annual energy simulation results showed that the addition of balcony thermal breaks was estimated to reduce the total annual energy consumption of the studied building, but only marginally (i.e., by less than 1%).

**Table 5.** The annual energy use intensity and energy cost for the case study building.

| Energy Use Categories | Energy Use Intensity (kWh/m$^2$) * | |
|---|---|---|
| | No Thermal Break | With Thermal Break |
| Space heating | 73.89 | 73.84 |
| Space cooling | 4.72 | 4.73 |
| Interior fans | 7.04 | 7.02 |
| Other ** | 64.73 | 64.72 |
| Total | 150.38 | 150.30 |

* EUI was calculated for the total residential area of 8,332 m$^2$ (99 units); ** Other energy use includes process energy not affected directly by thermal breaks (receptacle loads, interior lighting, domestic hot water, etc.)

3.3.2. Parametric Energy Study Results and Discussion

In order to evaluate the energy savings potential of balcony thermal breaks for a variety of more generic multi-family residential building designs, annual energy simulations were performed for each parameter combination described in Section 2.3.2 for the cases of uninsulated monolithic balconies and for the cases of thermally-broken balconies in the climate of Chicago, IL (O'Hare International Airport TMY3 weather file). Table 6 summarizes the annual building energy use intensity for all parameter combinations and the energy use reduction predicted by using the thermal breaks.

The use of balcony thermal breaks was predicted to yield total annual energy savings between 0.2% and 1.3%, depending on the building geometry, balcony geometry, and window-to-wall ratio (WWR). The greatest amount of savings was predicted for the 20-story building with continuous balconies and WWR of 100%. The smallest amount of savings was predicted for the 5-story building with single-unit balconies and WWR of 40%. In general, greater savings were predicted in the taller building models (20 stories versus 5 stories), in the models with continuous balconies, and in the models with higher WWR, all of which serve to increase the importance of the balcony slab connections in the envelope heat transfer. Additionally, the savings were more pronounced in cold months, with thermal breaks reducing annual space heating energy consumption by between 0.3% and 1.9% in Chicago's climate (Table 7). The magnitude of the predicted heating energy savings in all cases is lower than the previously predicted savings for multi-family buildings located in Canada estimated by Ge et al. (5%–11%) [4], Hardock et al. (7.3%) [5], and Baba et al. (7%–8%) [13].

Overall, this study demonstrates that balcony thermal breaks in the case study building in Chicago, IL are not expected to have a very large impact on annual building energy consumption and costs, although they were effective at regulating interior slab temperatures and increasing effective thermal resistance. Between both the case study building and the more generic building geometries, the predicted reductions in building energy use by incorporating thermal break products were smaller than the findings of several prior studies potentially due to the following:

- Previous studies may not have fully accounted for conduction through structural reinforcement in floor slabs, which increases its thermal bridging effect.
- Previous studies tend to only account for heating and cooling-related energy consumption alone, while this study included all energy use categories characteristic to the selected building type, including the energy used in heating, cooling, domestic hot water generation, interior lighting,

receptacles, elevators, fans, pumps, and heat rejection. The heating energy savings are less apparent when all energy use categories are accounted for.

- Previous studies have been conducted primarily in Canadian climates, which are characterized by colder winter temperatures.
- The assumptions for the building operation used in the energy models herein, which relied on usage profiles for interior lighting energy, miscellaneous receptacle loads, people occupancy, and domestic hot water (based on ASHRAE Research Project 1093-RP) [20], may have differed from the assumptions in previous studies.
- The thermal break product used in the case study building is one of the poorer-performing break products from the manufacturer. Additional simulations with better performing products demonstrated likely higher performance, albeit still with a small influence on annual energy consumption.

**Table 6.** The total annual energy use intensity (kWh/m$^2$) and energy use reduction.

| Balcony Geometry | Adjacent Exterior Wall | WWR | CT Balcony | TB Balcony | Savings % |
|---|---|---|---|---|---|
| **Number of Stories: 5** | | | | | |
| Single balconies | RSI Curtain wall | 100% | 302 | 301 | 0.48% |
| | RSI Insulated wall | 40% | 254 | 253 | 0.22% |
| Continuous balconies | RSI Curtain wall | 100% | 312 | 308 | 1.28% |
| | RSI Insulated wall | 40% | 259 | 257 | 0.65% |
| **Number of Stories: 20** | | | | | |
| Balcony Geometry | Adjacent Exterior Wall | WWR | CT Balcony | TB Balcony | Savings |
| Single balconies | RSI Curtain wall | 100% | 296 | 294 | 0.56% |
| | RSI Insulated wall | 40% | 248 | 247 | 0.30% |
| Continuous balconies | RSI Curtain wall | 100% | 306 | 302 | 1.33% |
| | RSI Insulated wall | 40% | 252 | 250 | 0.66% |

**Table 7.** The annual space heating energy use intensity (kWh/m$^2$) and energy use reduction.

| Balcony Geometry | Adjacent Exterior Wall | WWR | CT Balcony | TB Balcony | Savings % |
|---|---|---|---|---|---|
| **Number of Stories: 5** | | | | | |
| Single balconies | RSI Curtain wall | 100% | 204 | 202 | 0.69% |
| | RSI Ins. wall | 40% | 164 | 163 | 0.33% |
| Continuous balconies | RSI Curtain wall | 100% | 215 | 211 | 1.81% |
| | RSI Ins. wall | 40% | 169 | 167 | 0.97% |
| **Number of Stories: 20** | | | | | |
| Balcony Geometry | Adjacent Exterior Wall | WWR | CT Balcony | TB Balcony | Savings |
| Single balconies | RSI Curtain wall | 100% | 200 | 198 | 0.81% |
| | RSI Ins. wall | 40% | 159 | 158 | 0.46% |
| Continuous balconies | RSI Curtain wall | 100% | 212 | 208 | 1.88% |
| | RSI Ins. wall | 40% | 164 | 162 | 0.99% |

## 4. Conclusions

This study investigated the thermal effects and potential energy savings of installing balcony thermal breaks through a combination of measurements and models for a real case study building and for more generic building designs in Chicago, IL. The research findings can be summarized as follows:

1.  The addition of balcony thermal breaks allowed for a likely improvement in indoor thermal comfort in the residential units as the interior slab temperatures at the thermally-broken balconies were on average 1.2–1.6 °C lower in the summer and 1.1–1.3 °C higher in winter than those of the control units.

2.  The effective thermal resistance was estimated from the measured data and through a two-dimensional THERM analysis. Both approaches yielded reasonably similar results: The measured data averaged over night-time winter periods resulted in an estimated effective balcony thermal break RSI of ~0.12 K·m$^2$/W, while the THERM analysis resulted in an effective balcony thermal break RSI of ~0.29 K·m$^2$/W. Further, the THERM simulations demonstrated that the adjacent wall systems also affect the effective thermal resistance and their level of thermal performance (effective balcony thermal break RSI of ~0.37 K·m$^2$/W). Using better-performing balcony thermal breaks helps to increase the resistance of the balcony slab connection by an effective balcony thermal break RSI of ~0.44 K·m$^2$/W for the curtain wall and ~0.55 K·m$^2$/W for the insulated wall.

3.  The energy analysis conducted for the case study building predicted that the addition of balcony thermal breaks would only marginally reduce the annual energy consumption. Additional parametric energy analysis using more common geometries of a hypothetical multi-family residential building located in Chicago demonstrated that annual energy savings achievable by thermal break products greatly depend on a number of key building characteristics such as the building geometry, balcony geometry, and window-to-wall ratio (WWR). The annual heating energy consumption could be reduced by 0.3%–1.9%. The total annual building energy consumption could be reduced by 0.2%–1.3% based on a number of studied characteristics.

Overall, this work demonstrates that although balcony thermal breaks can indeed reduce thermal bridges and likely improve indoor thermal comfort by regulating interior slab temperatures, their predicted effect on annual building energy consumption and energy costs is relatively small. Thermal bridges at balcony connections typically comprise a small fraction of the total building envelope area (i.e., ~1–5% of the total exterior wall area) such that the use of balcony thermal breaks does not necessarily lead to a large improvement in energy performance. In fact, a building designer can more readily achieve energy performance improvements by reducing more prominent and abundant envelope thermal bridges, such as bridges due to the window and curtain wall frames, or by improving the thermal performance of the primary window and wall systems (adding more thermal insulation). Installing balcony thermal breaks should always be considered as part of the building envelope thermal improvements, alongside specifying high-performance window frames and well-detailed air barriers. However, the installation of balcony thermal breaks alone, without the inclusion of other envelope efficiency measures and excellent detailing of all envelope connections and interfaces, is not sufficient to solve the envelope thermal bridging problem.

**Author Contributions:** I.S. led the study design, data collection, data analysis, and preparation of the manuscript; B.S.(Brent Stephens) assisted in preparation for data collection, data analysis, and preparation of the manuscript; B.S.(Benjamin Skelton) conceived the study and assisted in study design and analysis.

**Funding:** This research received no external funding.

**Acknowledgments:** This study was conducted in collaboration between Cyclone Energy Group and the Department of Civil, Architectural, and Environmental Engineering at Illinois Institute of Technology. Funding for the study was provided by the architect of the case study building. Special appreciation goes to Haoran Zhao at Illinois Institute of Technology for his help in assembling the thermocouple wires prior to installation.

**Conflicts of Interest:** The authors declare no conflict of interest.

# Appendix A

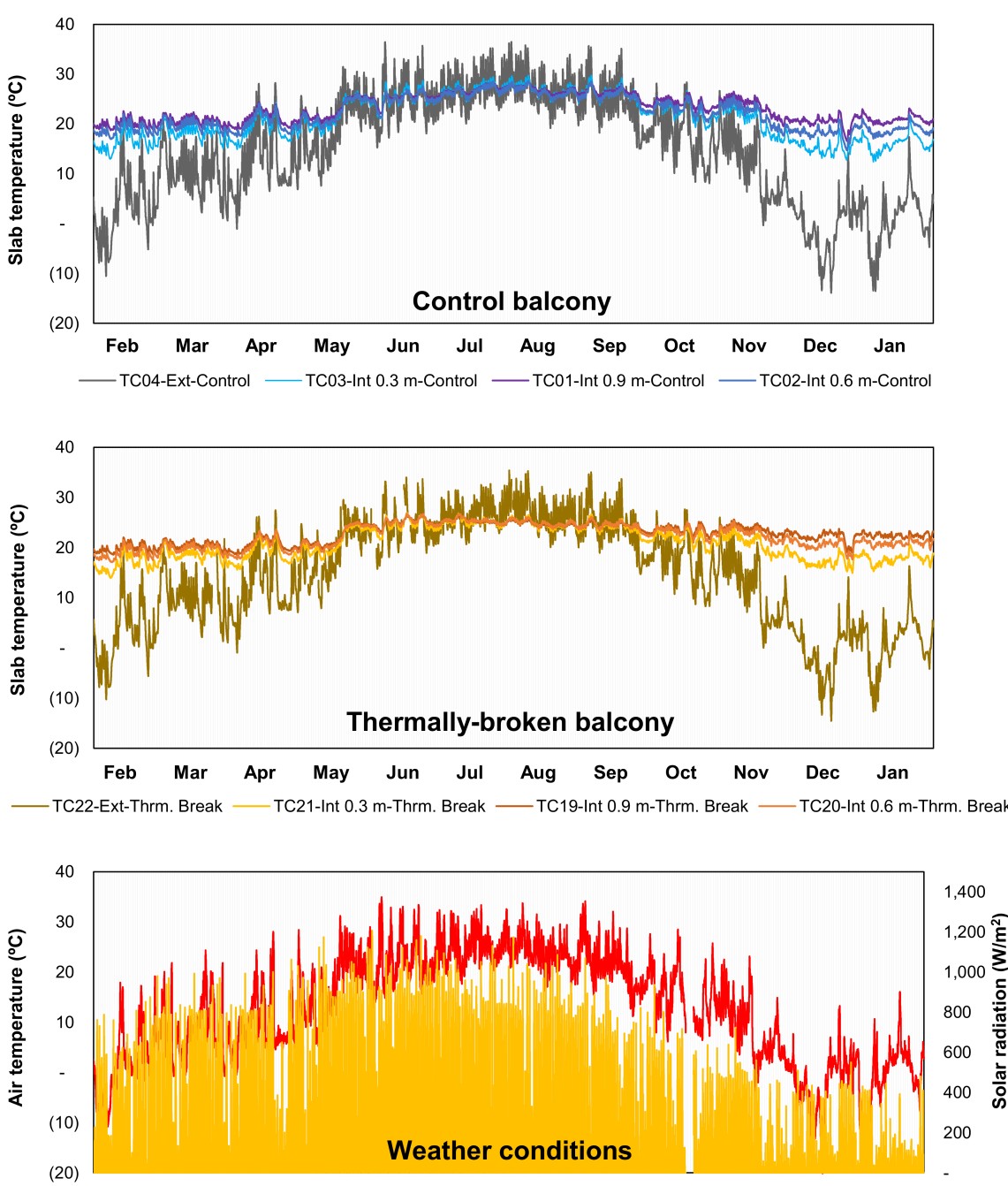

**Figure A1.** Experimental data for balcony pair #1.

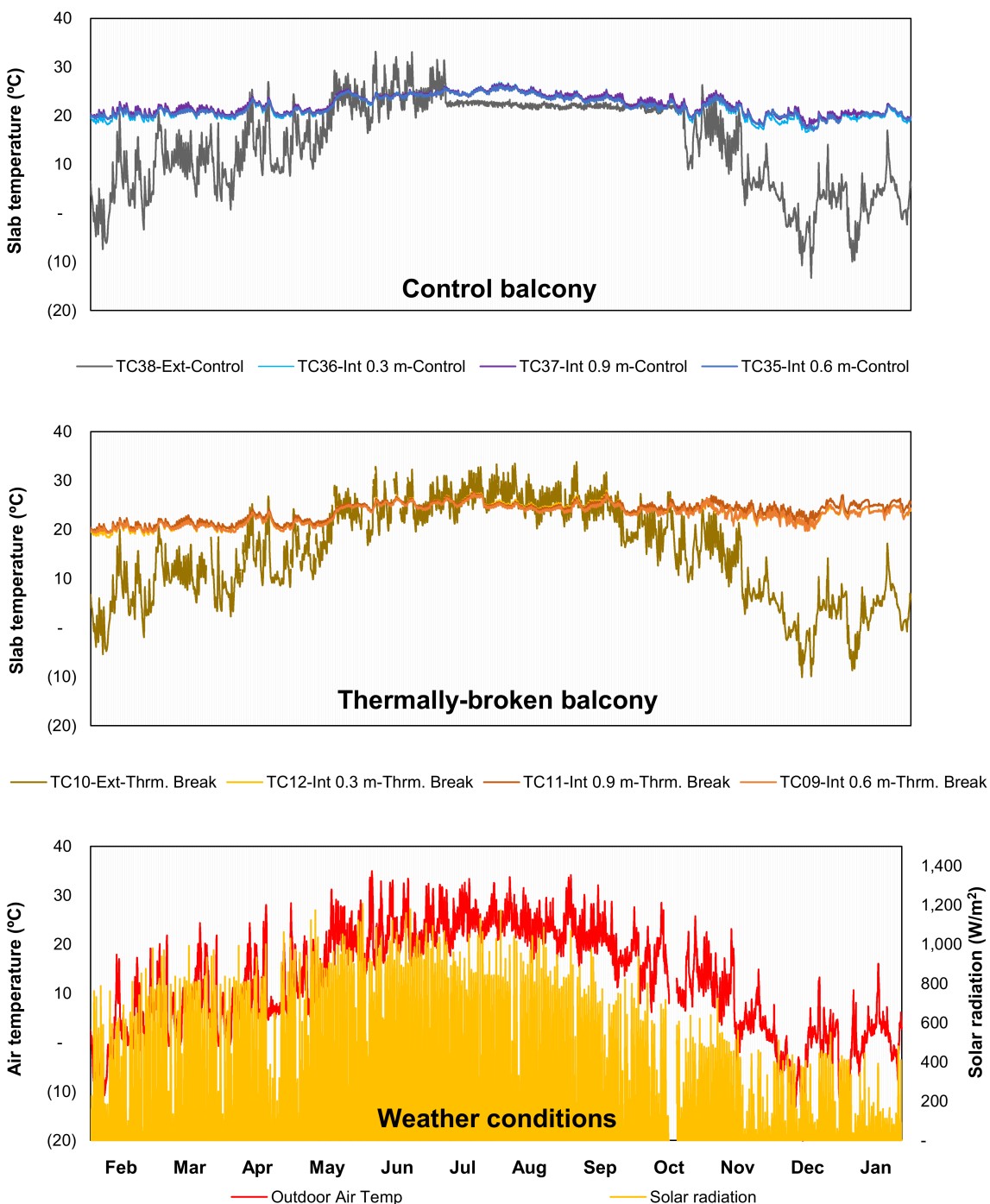

**Figure A2.** Experimental data for balcony pair #3.

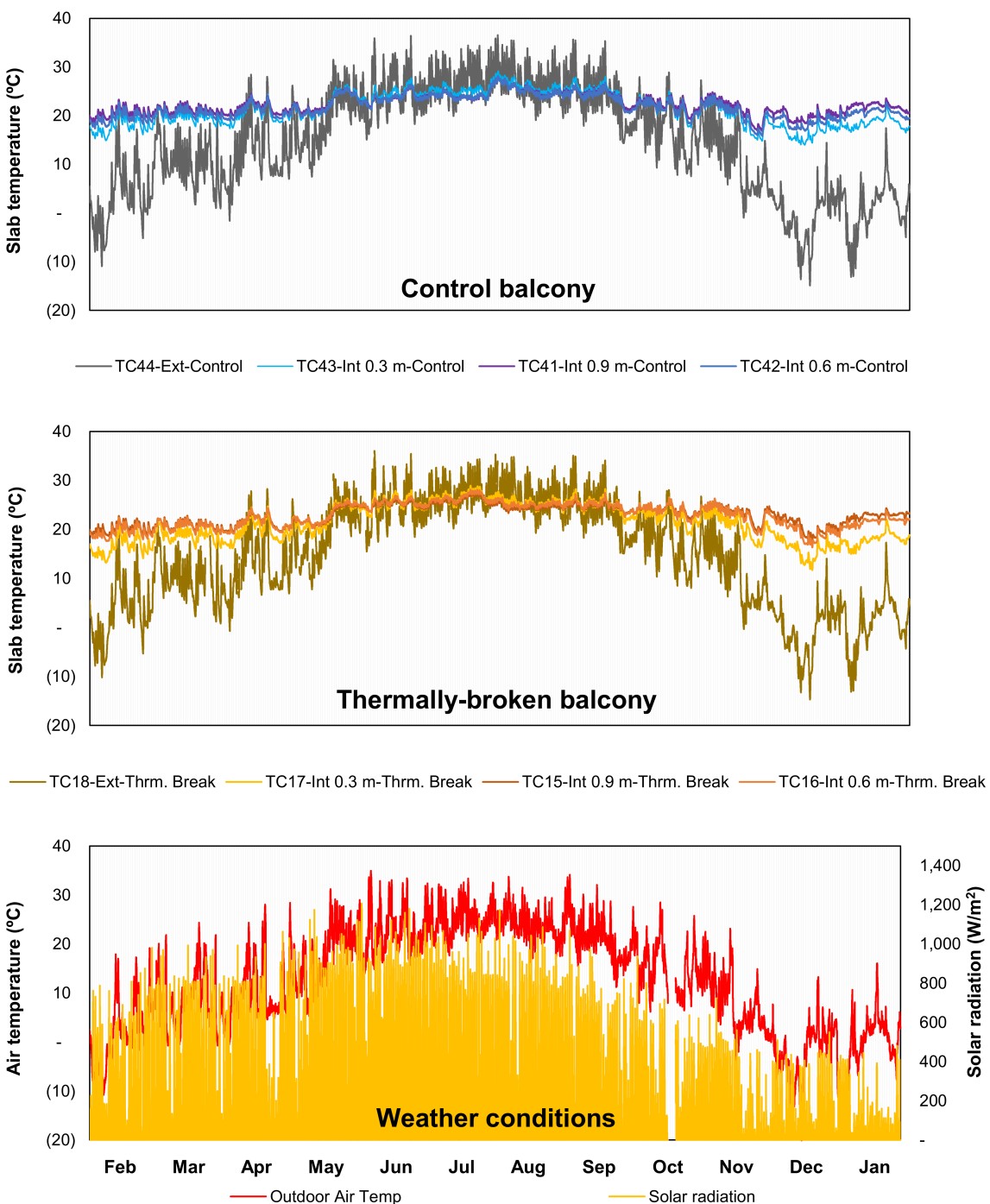

**Figure A3.** Experimental data for balcony pair #4.

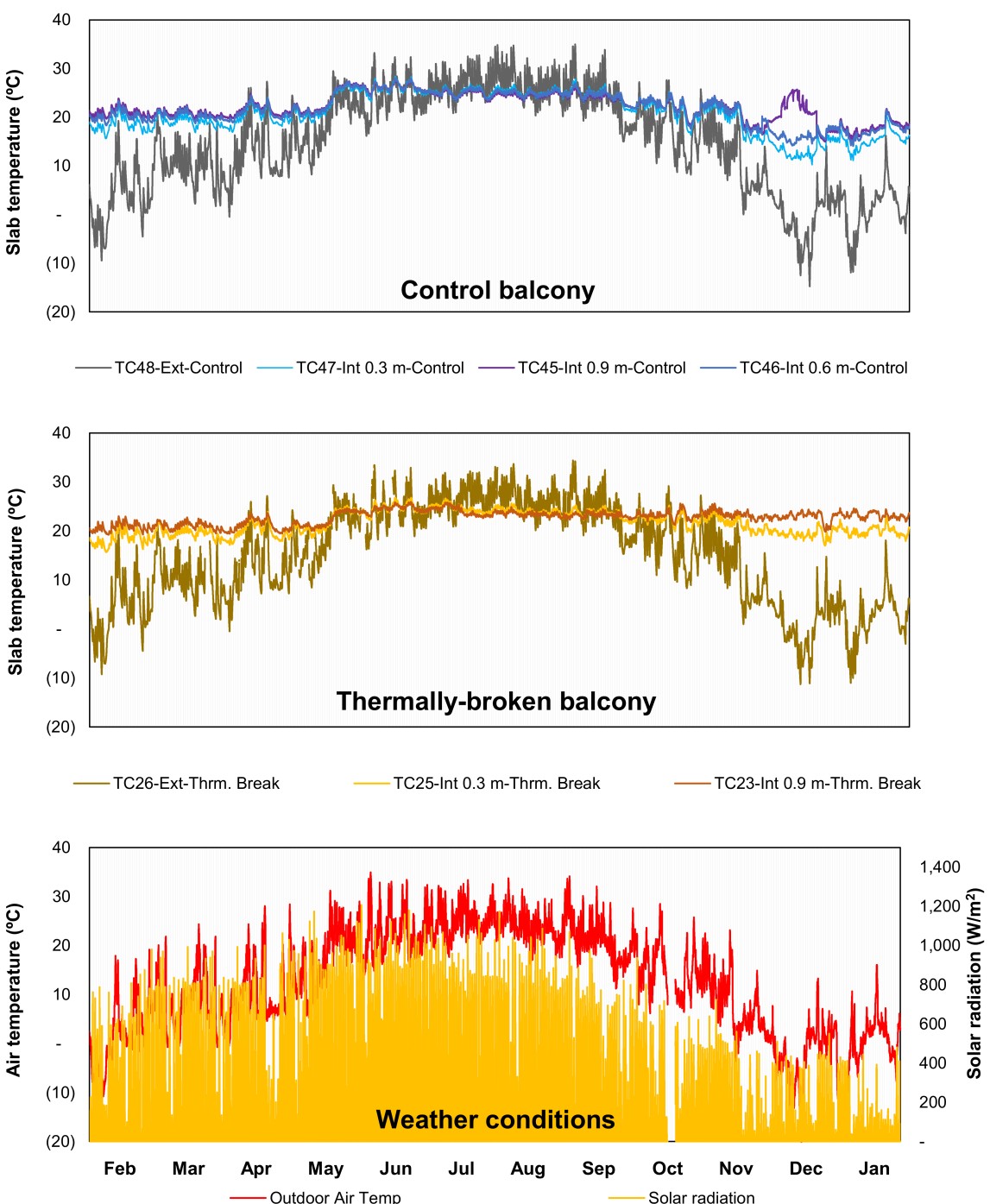

**Figure A4.** Experimental data for balcony pair #5.

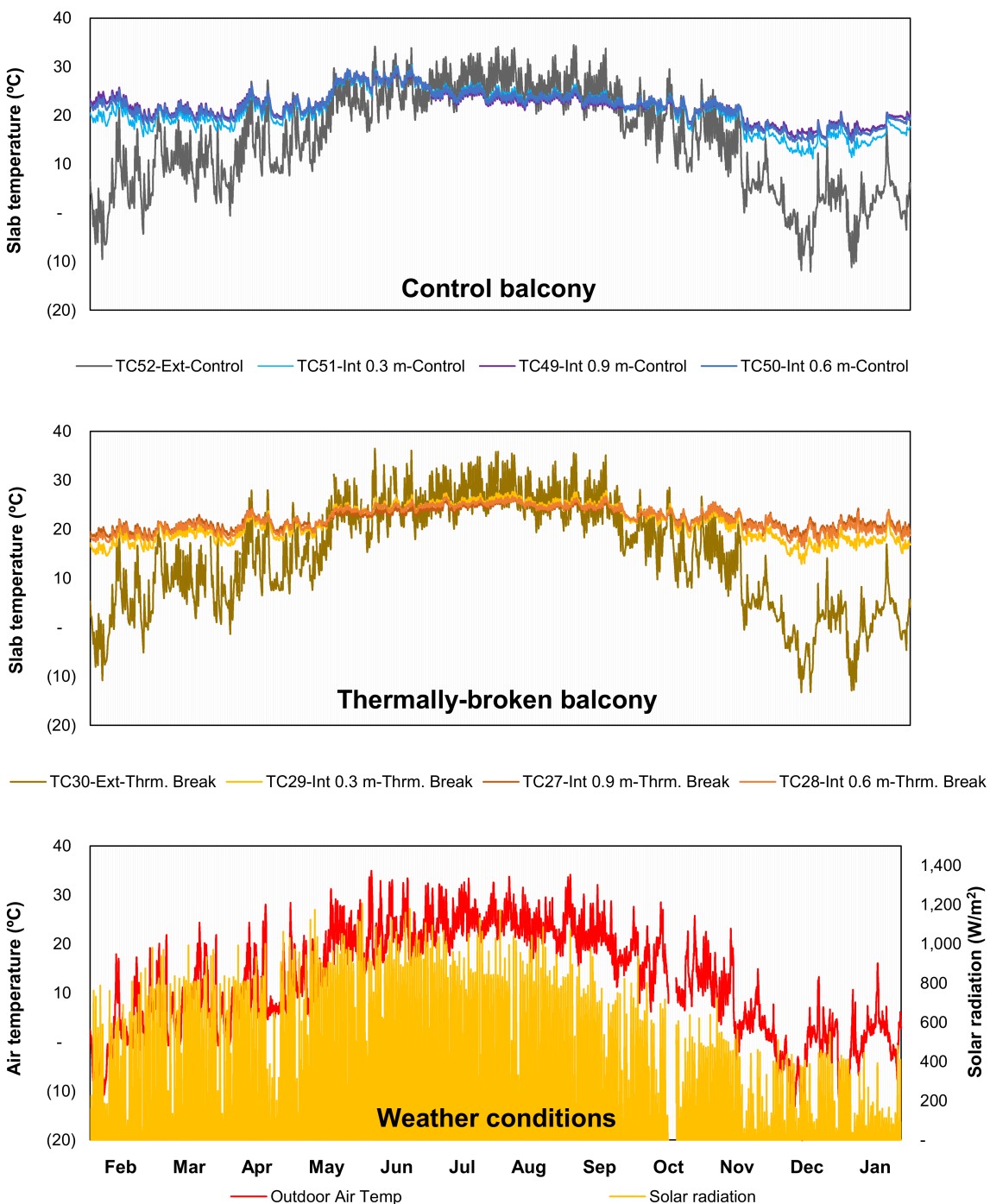

**Figure A5.** Experimental data for balcony pair #6.

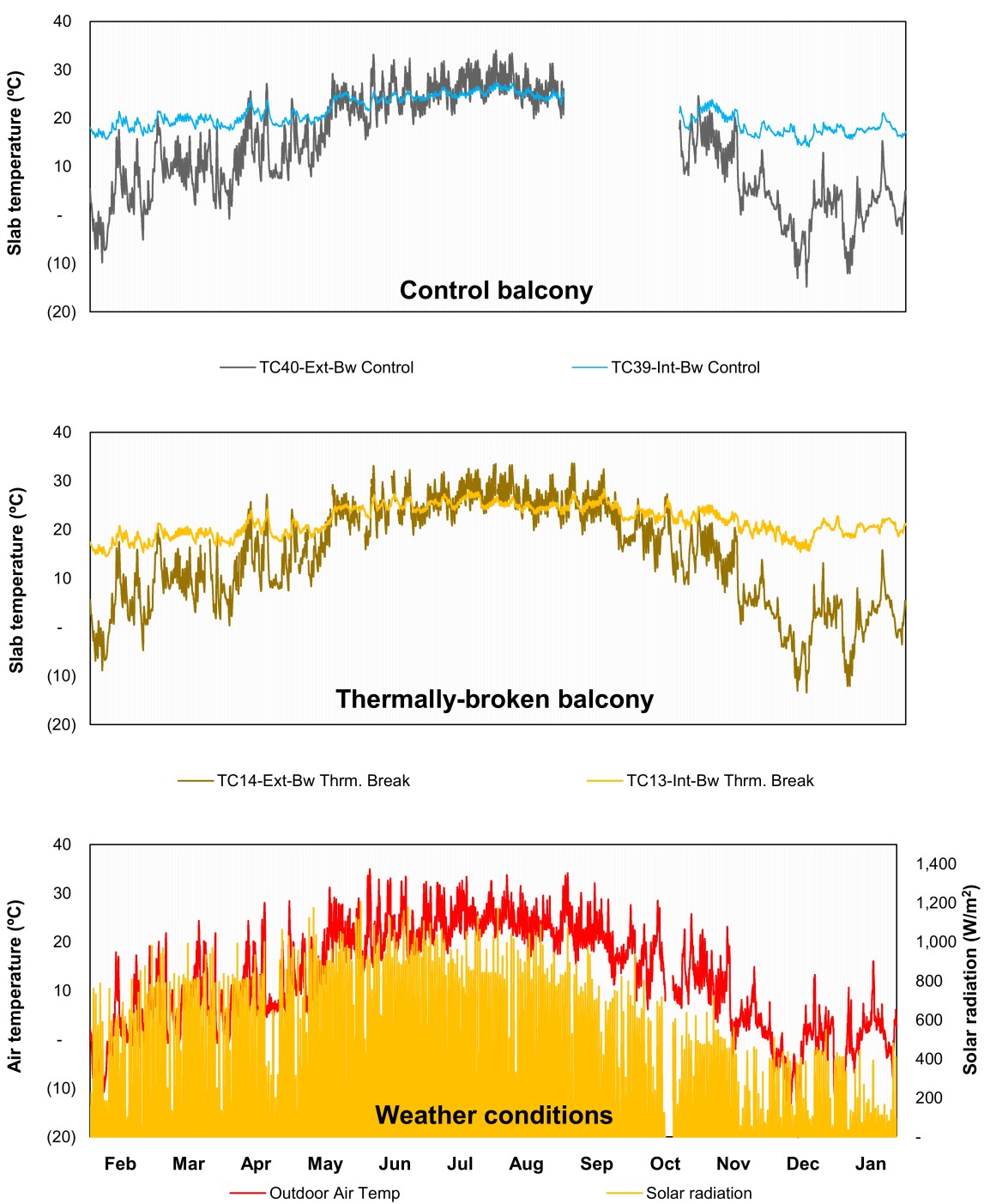

**Figure A6.** Experimental data for balcony pair #7.

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
