# Peer review of "The Effect of Balcony Thermal Breaks on Building Thermal and Energy Performance: Field Experiments and Energy Simulations in Chicago, IL"

_buildings, doi:10.3390/buildings9090190_

Round 1

Reviewer 1 Report

There are some things to be added that will vastly improve the paper.

While the conclusions indicate that the use of thermal breaks for balconies do not significantly impact on the energy use, it would be useful to cogitate about the use of the specific material that was used as the thermal break. There is an assumption that the material is integral and has undertaken its job. What if some other materials were used?

The paper does not describe what levels the thermal breaks were provided in? It appears there was only the one test case. What about shadowing from nearby structures/landscaping...this is not mentioned, but the presence of shadowing needs to be eliminated. 

The paper refers specifically to Canadian studies..what about European studies where thermal breaks are an essential part of their buildings due to the climatic conditions. Would be very useful to refer to studies from Europe to strengthen your case a bit more.

Finally, the conclusions need to be strengthened. It would be easy to take the approach that since thermal breaks do not make that much of a difference for balconies, why use it at all? It would be useful to stress the limitations of this study and consider future research opportunities based on this study.

Author Response

We appreciate these comments. Please find our response to all comments in blue below.

There are some things to be added that will vastly improve the paper.

While the conclusions indicate that the use of thermal breaks for balconies do not significantly impact on the energy use, it would be useful to cogitate about the use of the specific material that was used as the thermal break. There is an assumption that the material is integral and has undertaken its job. What if some other materials were used?

The most commonly used balcony thermal break insulation types are extruded polystyrene, expanded polystyrene, mineral fibers, and some proprietary foams, all of which have thermal conductivity in the range of 0.025-0.040 W/m·K. Thermal break manufacturers have products varying in insulation materials and thicknesses, as well as structural reinforcement, which may result in different levels of balcony thermal break insulation. For example, the thermal break product installed in the studied building has the lowest insulation level among the products offered by the same manufacturer. The overall thermal conductivity of this product was 0.294 W/m·K. A better product was available offering thermal conductivity up to 0.190 W/m·K (the difference is largely driven by the need for additional structural reinforcement for the case study, which reduces the overall insulation area of the thermal break product). However, the magnitude of differences between the better performing and lower performing products are not substantial.

To address this point, we have added some additional discussion and analysis about these better-performing products in the manuscript. Still, even with using a better-performing product in the analysis, the overall impact of balcony thermal breaks on building envelope thermal performance and annual energy savings remains small simply due to the fact that the envelope area improved by balcony thermal breaks occupies only 1%-5% of the total building envelope.

The paper does not describe what levels the thermal breaks were provided in? It appears there was only the one test case. What about shadowing from nearby structures/landscaping...this is not mentioned, but the presence of shadowing needs to be eliminated.

Four control and four thermally-broken balconies were studied. There were overall seven control/thermally-broken balcony connection pairs. See Section 2.1 for detailed discussion about the experimental setup. The temperature measurements were taken for one year and three representative months are shown in Figures S1-S6. However, since it was a real building and not everything goes according to the plan in real life (unlike it is in simulation studies), some sensors went bad (pair #2). For other sensors, we could not use the measured data because it did not correspond to our study’s criteria that the residential units must be occupied at the time of the study to yield realistic results. This left us with the measurements from sensor pair #1 as the most realistic data for further analysis.

To clarify details of the balconies further, all tested balconies are facing south and there are no adjacent structures that cast shadows onto the studied building throughout the year. This has been described in more detail in the text itself. The thermal analysis described in Section 3.2.1 is conducted using only the measurements when there is no solar radiation present (see discussion on page 11). 

The paper refers specifically to Canadian studies. What about European studies where thermal breaks are an essential part of their buildings due to the climatic conditions. Would be very useful to refer to studies from Europe to strengthen your case a bit more.

Some additional studies have been added to the reference list and cited in the manuscript, although it does not change the overall findings or relevance.

Finally, the conclusions need to be strengthened. It would be easy to take the approach that since thermal breaks do not make that much of a difference for balconies, why use it at all? It would be useful to stress the limitations of this study and consider future research opportunities based on this study.

Our conclusion is not that balcony thermal breaks should not be used. The conclusion is that installing balcony thermal breaks alone is not going to improve building thermal performance and yield great energy savings in and of itself. Balcony thermal breaks are just one of the measures that can lead to improved building envelope thermal resistance and thus improve energy performance. Balcony thermal breaks will work much better in conjunction with other important measures, such as high-performance window frames, good detailing of envelope connections and interfaces, and proper installation of air barriers (and our magnitude estimates suggest that for many buildings these other measures are probably more important to energy savings than thermal breaks alone). Overall, we intend to point out with these results that buildings are complex structures comprised of many energy-consuming systems, out of which building envelope is only one element. We think it is important to show the overall scale of magnitude of energy savings with balcony thermal breaks.

The most natural follow-up study would be what is the effect of reducing thermal bridging through window frames by using high-performance frames, especially since such thermal bridges can occupy much higher envelope area than balcony thermal bridges (depending on building geometry).

Reviewer 2 Report

The manuscript should be improved and, at least, the following changes should be carried out:

1. The novelty and originality should be highlighted.

2. Acronyms should be defined and used properly.

3. A more complete literature review is needed. The references are only 13 and too old.

4. The authors should avoid using “we”.

5. The main section are mixed and the manuscript should be re-structured. For instance, the authors should move some parts about materials and method from section 3 to section 2; or there are too many third-level headings (X.X.X.).

6. Section 3 should be called Results and discussion. This section is confusing and needs a deeper work.

Author Response

We appreciate these comments. Please find our response to all comments in blue below.

The manuscript should be improved and, at least, the following changes should be carried out:

The novelty and originality should be highlighted.

Multiple research studies of balcony thermal break effects were conducted using simulation-based approach or experimental measurements under controlled conditions using a hot-box apparatus. At the same time, there is little measured quantitative information available on how thermal breaks can affect in-situ building thermal performance and overall energy consumption. This research study seeks to fill this information gap by evaluating the thermal effect of balcony thermal breaks measuring an actual constructed building in the United States. We have attempted to articulate this novelty in the manuscript itself.

Acronyms should be defined and used properly.

All acronyms mentioned in the text have been spelled out:

ASHRAE - American Society of Heating, Refrigerating and Air-Conditioning Engineers TMY – typical meteorological year EER – energy efficiency ratio A more complete literature review is needed. The references are only 13 and too old.

Eight additional studies on thermal bridges and balcony thermal breaks have been added to the reference list and cited in the manuscript. We appreciate the encouragement to conduct a deeper literature review.

The authors should avoid using “we”.

“We” have been removed from the text of the manuscript.

The main section are mixed and the manuscript should be re-structured. For instance, the authors should move some parts about materials and method from section 3 to section 2; or there are too many third-level headings (X.X.X.).

We appreciate the comments to improve structure and readability. We have moved the 1-D calculation methods from Results (originally Section 3.2.1) to Methods (now Section 2.2.1). We also moved the THERM modeling methods from Results (originally 3.2.2) to Methods (now Section 2.2.2). We believe this more clearly structures the manuscript and also allows for maintaining the 3-level heading structure (which we believe helps segment the work).

Section 3 should be called Results and discussion. This section is confusing and needs a deeper work.

This section has been renamed based on the reviewer’s suggestion.

Round 2

Reviewer 2 Report

The manuscript has been improved, but the following changes should be carried out:

1. The word DATA is plural. Revise line 18.

2. The introduction should be improved. Moreover, the authors should avoid reference overkill. They should not use more than 3 references per sentence. If they need to use more, they will make sure the key idea of each single reference is mentioned. For instance, [1,2,4,7-9] in line 41 and [1,4-6,10-13] in line 55.

3. The authors should revise acronyms. For instance, ASHRAE is defined in line 233, but it was used in line 183.

4. The authors should revise unit system. For instance, K should be used in line 143, instead of ºK.

5. How were the experimental data used in the study? It is not clear in the manuscript.

6. Have the results and the new references been discussed in Section 3? It seems the authors have only added them without any criterium.

Author Response

We appreciate these comments. Please find our response to all comments in blue below.

The manuscript has been improved, but the following changes should be carried out:

The word DATA is plural. Revise line 18.

The verb conjugation has been corrected.

The introduction should be improved. Moreover, the authors should avoid reference overkill. They should not use more than 3 references per sentence. If they need to use more, they will make sure the key idea of each single reference is mentioned. For instance, [1,2,4,7-9] in line 41 and [1,4-6,10-13] in line 55.

The literature review and discussion in this submission has been improved and expanded.

The authors should revise acronyms. For instance, ASHRAE is defined in line 233, but it was used in line 183.

Per this comment, the acronym “ASHRAE” has been defined in the earlier section.

The authors should revise unit system. For instance, K should be used in line 143, instead of ºK.

This change has been made along with any other instances where ºK appeared. The remainder of the manuscript complies with SI units per the journal requirements.

How were the experimental data used in the study? It is not clear in the manuscript.

The data collected during the year-long measurement campaign are analyzed according to the method described in Section 2.2.1. The measurements are analyzed and presented in Sections 3.1 and 3.2.1 of this manuscript (they were used to calculate effective 1-D thermal resistance of the balconies with and without thermal breaks and to analyze the impact of thermal breaks on floor slab temperatures, which can affect comfort). These results are then tied to further heat transfer analysis in THERM (Sections 3.2.2 and 3.2.3) and to dynamic annual energy analysis in IES<VE> (Section 3.3).

Have the results and the new references been discussed in Section 3? It seems the authors have only added them without any criterium.

The results are discussed and compared to other studies at the end of Section 3.1 (improvement in indoor thermal comfort due to balcony thermal breaks), Section 3.2.3 (effective RSI of balcony thermal breaks), and Section 3.3.2 (heating energy and total energy savings with balcony thermal breaks). Indeed, we have incorporated a discussion of our results compared to those in the literature cited in the introduction.

Round 3

Reviewer 2 Report

The manuscript has been improved and only one change should be carried out:

What are the savings estimated by Baba el al. [13]? (Line 494)

Author Response

The annual heating energy savings reported by Baba et al. are 7%-8%. This information has been added to the manuscript per the reviewer's request.

Round 4

Reviewer 2 Report

The manuscript is suitable for publication in its present form